# Note on DNA Analysis and Redesigning Using Markov Chain

**DOI:** 10.3390/genes13030554

**Published:** 2022-03-21

**Authors:** Maciej Zakarczemny, Małgorzata Zajęcka

**Affiliations:** Department of Applied Mathematics, Faculty of Computer Science and Telecommunications, Cracow University of Technology (CUT), 24 Warszawska Street, 31-155 Cracow, Poland; malgorzata.zajecka@pk.edu.pl

**Keywords:** DNA, Markov chain, human gene, 92B05, 60J20

## Abstract

The paper contains a discussion on mathematical modifying and redesigning DNA with the use of Markov chains. We give a simple mathematical technique for overwriting missing parts of DNA. With a certain probability (without even knowing the function of the missing codon) we can find a synonymous codon, so that there is no frequency change in amino acid sequences of proteins. We use Markov Chain to analyze the dependencies in DNA sequence of the human gene Alpha 1,3-Galactosyltransfe rase 2. We include a theoretical introduction which facilitates the understanding of the paper for non-mathematicians, especially for biologists not familiar with the theory of Markov chains.

## 1. Introduction

### 1.1. Motivation and Methods

Modeling and analyzing DNA sequences using statistical methods have been a challenge for statisticians and biologists for many years. For many years, the most common approach was based on the theory of Markov chains. Well known simple models appeared in the mid-1980s in the papers by B.E. Blaisdell [1] and V. Brendel, J.S. Beckmann, E.N. Trifonov [2]. This was later followed by more advanced models developed to study different biological aspects of DNA (see [3,4,5]) that have been also using Markov chains (both homogeneous and non-homogeneous ones, possibly of higher orders). There are some statistical results showing that first-order Markov chains are not an adequate model for DNA sequences (see, e.g., [6]). Moreover, the latest comparison studies (see [7]) show that in general DNA sequencing by models based on even higher order Markov chains does not fit perfectly. Thus the latest research in DNA sequencing in bioinformatics is focusing on deep learning methods (see [8]). The methods presented in this paper, however, are local rather than statistical. We apply our model to DNA sequences less than 55 base pairs long, which is not enough for statistical methods. Markov chain theory is being applied to modeling music and literature. For example, a random song generated by a Markov chain based on some given piece of music, can achieve a similarity level comparable to this piece (see [9,10,11]). The question arises whether DNA can be handled similarly to a piece of music. Numerous attempts to write an understandable text as the realization of a low-order Markov chain have not proved successful. For high-order Markov chains, however, such a text becomes understandable as it contains complete sentences from the original text (see [12,13,14]). Therefore, the question whether Markov chains of any order are suitable to study DNA sequencing becomes the question about the complexity of the DNA structure. In this paper we show methods for filling in short gaps in DNA sequences. The results obtained by our method are then compared with the original DNA sequence.

### 1.2. Notations

We consider a probability space (Ω,F,P), where $\Omega =\{S_{1},S_{2},\ldots ,S_{n}\}$ denotes a finite state space, i.e., the set of all possible results of an experiment, F is a σ-field of events, and P:F→[0,1] denotes a probabilistic measure. By *random variable* we denote a function Z:Ω→R such that for any a∈R a preimage $Z^{-1}\left(\left(-\infty ,a\right]\right)$ is in the σ-field F. In our case a random variable will allow us to assign natural numbers to states from the state space Ω: let *E* be a single experiment with possible results forming Ω, if in *t*-th repetition (i.e., in time *t*) of experiment *E* we obtain result $S_{j}\in \Omega$, then we put $Z_{t}=j$, where $Z_{t}$ is a random variable with natural values.

### 1.3. Stochastic Matrices

**Definition** **1.**
*A matrix $\Pi ={\left(p_{ij}\right)}_{i,j\in \{1,\ldots ,n\}}$ is called stochastic if all $p_{ij}$ are non-negative and for any i we have $\sum_{j=1}^{n}p_{ij}=1$ (right stochastic matrix) or for any j we have $\sum_{i=1}^{n}p_{ij}=1$ (left stochastic matrix). A doubly stochastic matrix is both left stochastic and right stochastic. A vector with non-negative real elements is a stochastic vector if its elements sum to one.*


**Remark** **1.**
*Rows of right (columns of left) stochastic matrix are row (vertical) stochastic vectors.*


Using basic algebra one can prove the following remark.

**Remark** **2.**
*The product of two right (two left) stochastic matrices is a right (left) stochastic matrix.*


**Definition** **2.**
*A square matrix $A={\left(a_{ij}\right)}_{i,j\in \{1,\ldots ,n\}}$ is called irreducible if for any partition S∪T={1,…,n}, S∩T=⌀ there exists s∈S, t∈T such that $a_{st}\neq 0$.*


**Definition** **3.***Let A be a square matrix. An eigenvalue of A is a complex number λ such that det(A−λI)=0 (i.e., λ is a zero with multiplicity k≥1 of the* characteristic polynomial *$p_{A}\left(\lambda \right)=\det\left(A-\lambda I\right)$), where I denotes the identity matrix. The set of all eigenvalues is called spectrum.*

**Definition** **4.**
*The eigenvector of a of a square matrix A is a column vector v such that Av=λv, where λ is an eigenvalue of A. The left eigenvector of a square matrix A is a column vector w such that $w^{T}A=\lambda w^{T}$, where λ is an eigenvalue of A. Some authors define the left eigenvector as a row vector $w^{T}$.*


## 2. Markov Chains

A Markov chain is a sequence of random variables forming a probabilistic model describing a memoryless type of dependency: the future may depend only on the present and must be independent of the past.

### 2.1. Model

1.*E* is an experiment with possible results forming a finite set $\Omega =\{S_{1},S_{2},\ldots ,S_{n}\}$;2.S={1,2,…,n} is a state space associated with Ω;3.(Ω,F,P) is a probability space, where $F\subset 2^{\Omega }$ is a σ-field and *P* is a probabilistic measure P:F→[0,∞);4.$Z_{t}$ is a random variable defined as follows: if in *t*-th repetition of experiment *E* we obtain result $S_{j}\in \Omega$, then we put $Z_{t}=j\in S$.

**Definition** **5.***A sequence of random variables ${\left(Z_{t}\right)}_{t=0}^{\infty }$ with values in a state space S is a* Markov chain *if for all t∈N and all $j_{0},j_{1},\ldots ,j_{t}\in S$*
(1)$$P\left(Z_{t}=j_{t}|Z_{0}=j_{0},Z_{1}=j_{1},\ldots ,Z_{t-1}=j_{t-1}\right)=P\left(Z_{t}=j_{t}|Z_{t-1}=j_{t-1}\right)$$
*if only*
$$P(Z_{0}=j_{0},Z_{1}=j_{1},\ldots ,Z_{t-1}=j_{t-1})>0.$$

**Remark** **3.**
*In the general case a state space S can be an arbitrary countable subset of N (positive integers).*


**Remark** **4.**
*The Equation (Equation 1) is called Markov property and its right-hand side is called a transition operator i.e., the probability of moving from state $j_{t-1}$ to state $j_{t}$ in one step.*


**Definition** **6.**
*Let ${\left(Z_{t}\right)}_{t=0}^{\infty }$ be a Markov chain. For t≥1 we call a stochastic matrix $\Pi \left(t\right)={\left(p_{ij}\left(t\right)\right)}_{i,j\in S}$ a transition Matrix of a Markov chain $\left(Z_{t}\right)$ in time t if*

(2)
$$p_{ij}\left(t\right)=P\left(Z_{t}=j|Z_{t-1}=i\right)$$

*for all i such that $P(Z_{t-1}=i)>0$. Since the total of transition probability from one state to all other states must be equal to one, thus this matrix is a right stochastic matrix.*


**Definition** **7.**
*Let ${\left(Z_{t}\right)}_{t=0}^{\infty }$ be a Markov chain. If $P(Z_{t}=j_{t}|Z_{t-1}=j_{t-1})$ is independent of t, then we call this Markov chain homogeneous.*


**Remark** **5.**
*For our convenience we index states by t, but we remind the reader of the following property of homogeneous Markov chains*

$$P\left(Z_{t}=j_{t}|Z_{t-1}=j_{t-1}\right)=P\left(Z_{t+m}=j_{t}|Z_{t+m-1}=j_{t-1}\right),\;t,m\in N.$$



**Remark** **6.**
*If a Markov chain ${\left(Z_{t}\right)}_{t=0}^{\infty }$ is homogeneous then there exists a stochastic matrix $\Pi ={\left(p_{ij}\right)}_{i,j\in S}$ such that for all t≥1 transition matrix Π(t)=Π, where each value $p_{ij}$ is the probability of moving from state i to state j in one step.*


Now we are going to consider probabilities of moving from state to state in larger number of steps.

**Definition** **8.**
*Let ${\left(Z_{t}\right)}_{t=0}^{\infty }$ be a homogeneous Markov chain. For m≥1 we call a stochastic matrix $\Pi^{\left(m\right)}={\left(p_{ij}^{\left(m\right)}\right)}_{i,j\in S}$ a transition Matrix of a Markov chain $\left(Z_{t}\right)$ in m steps if*

(3)
$$p_{ij}^{\left(m\right)}=P\left(Z_{m}=j|Z_{0}=i\right)$$

*for all i such that $P(Z_{0}=i)>0$.*


**Theorem** **1.**
*Let ${\left(Z_{t}\right)}_{t=0}^{\infty }$ be a homogeneous Markov chain. Then $\Pi^{\left(m\right)}=\Pi^{m}$.*


**Proof.** For m=1 theorem is true because $\Pi^{\left(1\right)}=\Pi$. For m≥2 from the law of total probability we obtain
$$\begin{array}{c} p_{ij}^{\left(m\right)}=P\left(Z_{m}=j|Z_{0}=i\right)\\ =\sum_{s\in S}P\left(Z_{m-1}=s|Z_{0}=i\right)P\left(Z_{m}=j|Z_{m-1}=s\right)=\sum_{s\in S}p_{is}^{\left(m-1\right)}p_{sj}. \end{array}$$Hence $\Pi^{\left(m\right)}=\Pi^{\left(m-1\right)}\Pi$, thus $\Pi^{\left(m\right)}=\Pi^{m}$. □

**Example** **1**(Random walk on a complex plane). *Identify adenine with −i, cytosine with 1, guanine with −1, and thymine with i. Let ${\left(U_{n}\right)}_{n=0}^{\infty }$ be a sequence of independent random variables such that for any n≥1:*
$$P\left(U_{n}=1\right)=p_{1},\;P\left(U_{n}=i\right)=p_{2},\;P\left(U_{n}=-1\right)=p_{3},\;P\left(U_{n}=-i\right)=p_{4},$$
*where $p_{1}+p_{2}+p_{3}+p_{4}=1$, and $P(U_{0}=0)=1$.*
*For n≥1 define $Z_{n}=U_{0}+U_{1}+\ldots +U_{n}$ which is a random variable with values in state space of Gaussian integers Z[i]={a+bi:a,b∈Z}. Gaussian integers form a commutative ring and 2-dimensional integer lattice. We show that ${\left(Z_{n}\right)}_{n=0}^{\infty }$ is a Markov sequence.*

*Let $j_{0},j_{1},\ldots ,j_{n}\in Z\left[i\right]$ be such that $P(Z_{0}=j_{0},Z_{1}=j_{1},\ldots ,Z_{n-1}=j_{n-1})>0$. Then, from definition of $Z_{n}$,*

$$\begin{array}{c} P(Z_{n}=j_{n}|Z_{0}=j_{0},Z_{1}=j_{1},\ldots ,Z_{n-1}=j_{n-1})\\ =\frac{P(Z_{0}=j_{0},Z_{1}=j_{1},\ldots ,Z_{n-1}=j_{n-1},Z_{n}=j_{n})}{P(Z_{0}=j_{0},Z_{1}=j_{1},\ldots ,Z_{n-1}=j_{n-1})}\\ =\frac{P(U_{0}=j_{0},U_{1}=j_{1}-j_{0},\ldots ,U_{n-1}=j_{n-1}-j_{n-2},U_{n}=j_{n}-j_{n-1})}{P(U_{0}=j_{0},U_{1}=j_{1}-j_{0},\ldots ,U_{n-1}=j_{n-1}-j_{n-2})}\\ =\frac{P\left(U_{0}=j_{0}\right)P\left(U_{1}=j_{1}-j_{0}\right)\cdot \ldots \cdot P\left(U_{n-1}=j_{n-1}-j_{n-2}\right)P\left(U_{n}=j_{n}-j_{n-1}\right)}{P\left(U_{0}=j_{0}\right)P\left(U_{1}=j_{1}-j_{0}\right)\cdot \ldots \cdot P\left(U_{n-1}=j_{n-1}-j_{n-2}\right)}\\ =P(U_{n}=j_{n}-j_{n-1}) \end{array}$$

*because variables ${\left(U_{n}\right)}_{n=0}^{\infty }$ were independent. Observe that variables $U_{n}$ and $Z_{n-1}$ are also independent, thus*

$$\begin{array}{c} P\left(U_{n}=j_{n}-j_{n-1}\right)=\frac{P\left(U_{n}=j_{n}-j_{n-1}\right)P\left(Z_{n-1}=j_{n-1}\right)}{P(Z_{n-1}=j_{n-1})}\\ =\frac{P(Z_{n-1}=j_{n-1},U_{n}=j_{n}-j_{n-1})}{P(Z_{n-1}=j_{n-1})}=\frac{P(Z_{n-1}=j_{n-1},Z_{n}=j_{n})}{P(Z_{n-1}=j_{n-1})}\\ =P(Z_{n}=j_{n}|Z_{n-1}=j_{n-1}). \end{array}$$


*One can show that $E(|Z_{n}|)$—the expected translation distance after n steps is of order $\sqrt{n}$, more precise, $\lim_{n\to \infty }\frac{E(|Z_{n}|)}{\sqrt{n}}=\sqrt{\frac{\pi }{2}}$. With our identification of adenine (−i), cytosine (1), guanine (−1), and thymine (i) using Markov chain ${\left(Z_{n}\right)}_{n=0}^{\infty }$ we can consider probability that from n-th to m-th place in DNA strand, n<m, number of adenine equals number of thymine and, simultaneously, number of cytosine equals number of guanine. This situation means that our random walk made a loop, that is $Z_{n}=Z_{m}$ (see Figure 1). If one needs to research other pairwise equalities it suffices to change the identification. Denote $Z\left[n_{1},n_{2}\right]=\left\{Z_{n}:n_{1}\leq n\leq n_{2}\right\}.$ One can show that for all k≥2 we have P(Z[0,k]∩Z[2k,3k]≠⌀)>0. That means that with positive probability there exists n∈[0,k] and m∈[2k,3k] such that $Z_{n}=Z_{m}$, i.e., we have a loop.*


### 2.2. Classification of States and Chains

In this subsection, we will give some mathematical background and also state some well known results, see [7,15].

**Definition** **9.**
*A state i is called accessible from state j if there exists n≥0 such that $P(Z_{n}=i|Z_{0}=j)>0$. If state i is accessible from state j and vice versa we say that states i and j communicate.*


Observe that communication is an equivalence relation that divides states into equivalence classes called *communicating classes*.

**Definition** **10.**
*A Markov chain is called irreducible if its state space forms a single communicating class.*


In other words, in irreducible Markov chain it is possible to get from any state to any state (every two states communicate).

**Definition** **11.**
*A state i is called inessential if there exists a state j and n≥1 such that $P(Z_{n}=j|Z_{0}=i)>0$ and $P(Z_{k}=i|Z_{0}=j)=0$ for any k≥0. A state is essential if it is not inessential.*


Denote $f_{ij}^{\left(n\right)}=P\left(Z_{n}=j,Z_{n-1}\neq j,\ldots ,Z_{1}\neq j|Z_{0}=i\right)$, $F_{ij}=\sum_{n=1}^{\infty }f_{ij}^{\left(n\right)}$, $P_{i}=\sum_{n=1}^{\infty }p_{ii}^{\left(n\right)}$ and define $N_{i}=\sum_{n=1}^{\infty }1_{\left\{Z_{n}=i\right\}}$, $M_{i}=\sum_{n=1}^{\infty }1_{\left\{T_{ii}\geq n\right\}},$ where $T_{ij}=\inf\left\{n\in N:Z_{n}=j\right\}$ when $Z_{0}=i$. Then $f_{ij}^{\left(n\right)}$ is a probability that we access state *j* from state *i* exactly in *n* steps, $F_{ij}$ is a probability that we ever access state *j* from state *i*, $P_{i}$ a trace of the transition matrix in *n* steps, $N_{i}$ is a random variable that counts how many times state *i* is accessed and $M_{i}$ is a random variable that counts how many steps are needed to reappear in state *i* for the first time.

**Remark** **7.**

$$P_{i}=\sum_{n=1}^{\infty }P\left(Z_{n}=i|Z_{0}=i\right)=\sum_{n=1}^{\infty }E\left(1_{\left\{Z_{n}=i\right\}}|Z_{0}=i\right)=E\left(N_{i}|Z_{0}=i\right),$$

*where E denotes an expected value. Thus $P_{i}$ is an average time a Markov chain is in state i (averagely how many times Markov chain is in state i). If we define $\mu_{i}=\sum_{n=1}^{\infty }P\left(T_{ii}\geq n|Z_{0}=i\right)$ then $\mu_{i}=E\left(M_{i}|Z_{0}=i\right)$. Thus $\mu_{i}$ is an average number of steps needed to reappear in state i.*


**Definition** **12.**
*A state i is called recurrent if $F_{ii}=1$. If $F_{ii}<0$ then state i is called transient.*


We will need the following result.

**Theorem** **2**(see [15]). *(a)* *A state i is recurrent if and only if $P(N_{i}=\infty |Z_{0}=i)=1$.**(b)* *A state i is transient if and only if $P(N_{j}<\infty |Z_{0}=i)=1$.*

**Theorem** **3.**
*(a)* 
*A state i is transient if and only if $P_{i}<\infty$.*
*(b)* 
*A state i is recurrent if and only if $P_{i}=\infty$.*



**Proof.** Note that for all states i,j and natural *n* we have
(4)$$p_{ij}^{\left(n\right)}=\sum_{m=1}^{n}f_{ij}^{\left(m\right)}p_{jj}^{\left(n-m\right)}.$$It follows from the fact that accessing state *j* from state *i* after *n* steps means that we access state *j* for the first time after exactly *m* steps (for some m≤n) and then after next n−m steps we return to it (perhaps reaching state *j* a few times on the way). Thus we have
$$\begin{array}{cc} \sum_{k=1}^{n}p_{ii}^{\left(k\right)} & =\sum_{k=1}^{n}\sum_{m=0}^{k-1}f_{ii}^{\left(k-m\right)}p_{ii}^{\left(m\right)}=\sum_{m=0}^{n-1}p_{ii}^{\left(m\right)}\sum_{k=m+1}^{n}f_{ii}^{\left(k-m\right)}\\ \; & \leq \sum_{m=0}^{n}p_{ii}^{\left(m\right)}F_{ii}=F_{ii}+F_{ii}\sum_{m=1}^{n}p_{ii}^{\left(m\right)}. \end{array}$$Hence we get an inequality
(5)$$\left(1-F_{ii}\right)\sum_{m=1}^{n}p_{ii}^{\left(m\right)}\leq F_{ii}.$$Since *n* is arbitrary, as *n* tends to infinity we obtain
(6)$$\left(1-F_{ii}\right)P_{i}\leq F_{ii}.$$Assume that state *i* is transient. Then from (Equation 6) we obtain $P_{i}<\infty$. On the other hand if $P_{i}<\infty$ then $E(N_{i}|Z_{0}=i)<\infty$, thus $P(N_{i}=\infty |Z_{0}=i)=0$. From Theorem 2 state *i* is transient. If *i* is recurrent, then we must have $P_{i}=\infty$ (otherwise see proof of point (a)). Now assume $P_{i}=\infty$. If $F_{ii}<1$ then $\left(1-F_{ii}\right)P_{i}$ is unbounded and from (Equation 6) we get a contradiction. □

**Remark** **8.**
*In irreducible Markov chain either all states are recurrent or all states are transient. Thus we call an irreducible Markov chain recurrent or transient, depending on type of states.*


**Remark** **9.**
*One can show that for a finite Markov chain (chain with a finite state space) a state is inessential if and only if it is transient, thus a state is essential if and only if it is recurrent.*


**Remark** **10.**
*In the case of gene A3GALT2 each state is essential (because all entries in transition matrix are nonzero), thus from Remark 9 each state is recurrent. This also can be shown using properties of transition matrix: evaluate $p_{ii}^{\left(n\right)}$ from the matrix $\Pi^{n}$. Then $P_{i}=\infty$ for each i∈{a,c,g,t}. From Theorem 3 we once again obtain that each state is recurrent.*


**Definition** **13.**
*A state i is called null-recurrent if $\lim_{n\to \infty }p_{ii}^{\left(n\right)}=0$. A state which is not null-recurrent is called positive recurrent.*


**Definition** **14.**
*A state i is called periodic with period $d_{i}$ if $d_{i}=\mathrm{GCD}\left\{n>0:p_{ii}^{\left(n\right)}>0\right\}>1$ (if for all n>0 we have $p_{ii}^{\left(n\right)}=0$ then we put $d_{i}=\infty$). If $d_{i}=1$ state i is called aperiodic.*


**Definition** **15.**
*A state which is aperiodic, recurrent, and positive recurrent is called ergodic.*


**Remark** **11.**
*In case of our matrix Π all states are ergodic.*


For irreducible matrices we have the following property.

**Theorem** **4.**
*A Markov chain is irreducible if and only if for all j∈{1,…,n} there exists a limit*

$$\lim_{t\to \infty }p_{ij}^{\left(t\right)}=p_{j},\;i,j\in \left\{1,\ldots ,n\right\},$$

*independent of i, where $p_{j}$, j∈{1,…,n}, form a unique solution of the following system of equations*

$$\left\{\begin{array}{c} \sum_{i=1}^{n}p_{i}p_{ij}=p_{j},\;j\in \left\{1,\ldots ,n\right\}\\ \sum_{j=1}^{n}p_{j}=1. \end{array}\right.$$



A special case of irreducible Markov chain is a regular Markov chain.

**Definition** **16.**
*A irreducible Markov chain is called regular if there exists k∈N such that all entries of the matrix $\Pi^{k}$ are positive. In other words there exists k∈N such that from any state we can reach any state in exactly k steps.*


**Definition** **17.***Let Π be a transition matrix of a Markov chain. A stationary probability vector is a stochastic vector (see Definition 1) such that π=πΠ. In other words π is a stochastic eigenvector associated with eigenvalue λ=1 of matrix* Π.

**Theorem** **5**(see [15]). *Let $\Pi^{\left(m\right)}={\left(p_{ij}^{\left(m\right)}\right)}_{i,j\in S}$ be a transition matrix in m steps of an irreducible aperiodic Markov chain $\left(Z_{n}\right)$ with finite state space S. Then*
*(i)* *for any i,j∈S there exists a limit $\lim_{m\to \infty }p_{ij}^{\left(m\right)}=\pi_{j}$, where $\pi_{j}>0$,**(ii)* *Markov chain $\left(Z_{n}\right)$ is recurrent,**(iii)* *a vector $\pi ={\left(\pi_{j}\right)}_{j\in S}$ is a unique stationary probability vector, moreover,**$\pi_{j}=\frac{1}{\mu_{j}}$ where $\mu_{j}$ is an average number of steps needed to reappear in state j.*

For regular Markov chains we have the following result.

**Theorem** **6.**
*Let Π be a transition matrix of an irreducible aperiodic Markov chain with finite state space. Then matrix $\Pi^{\left(m\right)}$ converges to a positive stochastic matrix W such that if π is a row of matrix W, then π=πΠ.*


### 2.3. Analysis of Alpha 1,3-Galactosyltransferase 2

We show that a time homogeneous Markov chain is an appropriate simple model of Alpha 1,3-Galactosyltransferase 2 (A3GALT2). We use transition matrices as a criterion for identifying similarities in structure of this particular gene. A3GALT2 is a Protein Coding gene (a region of DNA) located in chromosome 1, position 33,306,766, consisting of 14,333 bases [16].

Let $\Omega =\{S_{1},S_{2},S_{3},S_{4}\}$, where $S_{1}=A,\;S_{2}=C,\;S_{3}=G,\;S_{4}=T$, and S={1,2,3,4} be a corresponding state space. We form a stochastic matrix (Equation 7) as follows. For example we would like to know how probable is that after adenine (A) occurs cytosine (C). We count all occurrences of a pair AC in gene A3GALT2 and divide it by number of all occurring pairs which start from A. Number of all such pairs is equal to number of occurrences of A provided that A is not the last nucleotid base in gene A3GALT2.
(7)$$\Pi =\begin{array}{cc} \; & \begin{array}{cccc} A\; & \;C & \;G & \;T \end{array}\\ \begin{array}{c} \;A\\ \;C\\ \;G\\ \;\;T \end{array}\; & \;\left(\begin{array}{cccc} \frac{769}{3195} & \frac{745}{3195} & \frac{1093}{3195} & \frac{588}{3195}\\ \frac{1140}{4052} & \frac{1392}{4052} & \frac{350}{4052} & \frac{1170}{4052}\\ \frac{811}{3715} & \frac{934}{3715} & \frac{1254}{3715} & \frac{716}{3715}\\ \frac{475}{3370} & \frac{980}{3370} & \frac{1018}{3370} & \frac{897}{3370} \end{array}\right) \end{array}$$

Note that because the last nucleotide base in gene A3GALT2 is T, in the denominator in last row we have 3370 instead of 3371.

**Remark** **12.**
*One can pose a question of biological interpretation of matrix Π. Does the occurrences of nucleotid bases can be used to identify a specific gene or, in general case, to identify an individual?*


Because all entries of matrix Π are positive, in the case of gene A3GALT2 a suitable Markov chain is irreducible (see Definition 10) and each $d_{i}=1$ for i∈S thus our chain is aperiodic (see Definition 14). From Theorem 5 for any i,j∈S there exists a limit $\lim_{m\to \infty }p_{ij}^{\left(m\right)}=\pi_{j}>0$, suitable Markov chain is recurrent, $\pi ={\left(\pi_{j}\right)}_{j\in S}$ is a unique stationary probability vector and $\pi_{j}=\frac{1}{\mu_{j}}$ where $\mu_{j}$ is an average number of steps needed to reappear in state *j*.

**Remark** **13.**
*The stationary probability vector of matrix Π is*

π=(0.22292,0.28265,0.25923,0.23521).



**Remark** **14.**
*Note that if nucleotide bases in gene A3GALT2 are a good estimation of a possible sequence of values of a Markov chain, then from Remark 13 probability of occurrence of a should be 0.22292, c: 0.28265, g: 0.25923 and t: 0.23521. Comparing this with computed probabilities from Table 1 (a: 0.222912, c: 0.282704, g: 0.259192, t: 0.235192) we see that they are correct up to the fourth decimal place.*


**Corollary** **1.**
*From Remark 13 we can compute approximate values of $\mu_{j}$: $\mu_{1}\approx 4.486$, $\mu_{2}\approx 3.538$, $\mu_{3}\approx 3.858$, $\mu_{4}\approx 4.252$ which means that an average number of steps needed for each nucleotide base to reappear in our gene is approximately 4 for all bases. Thus we conclude that bases are uniformly distributed in gene A3GALT2 which means that they appear to be random and disorganized.*


All of the above considerations can be repeated for pairs of bases, see Table 2.

## 3. The Markov Process Model of Nucleotide Substitution

We assume that nucleotide substitution is follow a homogeneous Markov process. We take $\Omega =\{S_{1},S_{2},S_{3},S_{4}\}$, where $S_{1}=A,\;S_{2}=C,\;S_{3}=G,\;S_{4}=T$. Let S={1,2,3,4} be a corresponding state space. Let $P\left(t\right)=\{P_{\mu \nu }\left(t\right)\}$ is a matrix of transition probabilities in time t. We assume that $P^{\prime }\left(t\right)=QP\left(t\right)$ where $Q=\{Q_{\mu \nu }\}$ is the rate matrix of the process.

**Remark** **15.**
*Figure 2, Figure 3, Figure 4 and Figure 5 show four situations in which the sequence starts with adenine, cytosine, guanine, and thymine, respectively. In each case, the probabilities of occurence of a given base stabilize. Finally, they converge to the probabilities forming a stationary probability vector. Note that, as predicted by Corollary 1, in each case stabilization is achieved after about four steps.*


## 4. Application in DNA Sequencing, Redesigned DNA

The meaning of DNA sequencing is here deemed to cover all methods used to determine the order of nucleotides along a DNA strand. The objective of this section is to present an example of applying Markov chains to complete short fragments of a DNA strand. Markov chains have been applied as mathematical models of real-life processes. Such real-life dynamical systems, examined with Markov-chain method, include

-queues of passengers arriving at an airport,-currency exchange rates or-animal population dynamics.

Markov chains are also applied to build algorithms calculating the PageRank value for a website (see [17]). The website PageRank value reflects the probability that a random internet surfer will land on this webpage upon clicking a link. Markov processes of various orders are used to model DNA sequencing (see also [6]). A Markov process of order *m* is one for which the probability of any event depends exclusively on the m preceding events. Statistically, DNA does not have the features of a first-order Markov chain. Higher-order models have been proposed for analyzing interrelations within a DNA sequence, see [6]. Statistical tests used in [6] properly determined the order of the Markov chain being tested for sequences of length $2^{9}$ base pairs or higher. Nevertheless, the authors consider it important to present the method described below for local problems, that is for very short DNA sequences (54 base pairs in our example). We believe it is worth examining the results of local DNA completion based on the properties of first-order Markov chains, when the length of a DNA sequence is less than $2^{9}$ base pairs. The method presented below is a simple, local tool for completing short DNA segments. The method has also educational value. Moreover, instead of analyzing a DNA sequence of the five unit nucleotides, we may use first-order Markov chains to analyze codons or, more precisely, amino acids encoded with those codons. It is worth recalling here some basic facts and conventions:-Codon is a sequence of three nucleotides (a triplet) occurring in mRNA, a unit encoding a specific amino acid during protein synthesis;-Proteins are built of 20 different amino acids;-The sequence of amino acids in a protein exactly follows the sequence of the relevant codons in mRNA;-Most amino acids are encoded in several ways (with different codons, which, however, differ from one another usually on the third place in the triplet only); owing to this, certain changes in the genetic information (mutations) do not affect the amino acid sequence;-There are 61 codons encoding amino acids and 3 non-encoding codons (they are STOP codons: UAG, UAA, UGA); all in all: $4^{3}$ various triplets;-The AUG codon, read as the first one in mRNA by a ribosome during protein synthesis is known as the initiation or start codon;-Since a mutation of a single nucleotide changes a single amino acid, the genetic code has to be read as non-overlapping, i.e., any given codon may be followed by any other codon;-To get the form typical of DNA, each U in an mRNA codon should be replaced by T; for instance, TAA is the DNA equivalent of the mRNA codon UAA;-In the case of a sequence of amino acids, understood as resulting from first-order Markov process action, the transition matrix is a square matrix of degree at most 21. One state is reserved for the three STOP codons, which do not encode amino acids.

**Example** **2.**
*Let us consider the human SATB1 gene, which, as research has revealed, is a major growth factor for breast cancer, see [18]. Let us generate a DNA sequence based on SATB1 (this gene is on chromosome 3, locus p23, on the minus strand). The table below sets forth a 54-base-pair-long segment of SATB1, position 18,389,139. Data is sourced from website [19], accessed upon entering human gene SATB1.*

*The relevant state space comprises four nucleotides: $\Omega_{1}=\left\{A,C,G,T\right\}.$ The corresponding amino acid sequence is:*

ValLysArgLeuSerAspLysAsnLysSerSerLeuSTOPGlnLeuCysCysSTOP.


*We give another sequence in Table 3, as a variation of the method consists in examining the sequence of amino acids and not the DNA sequence of base pairs. For the sequence of the DNA segment under this analysis, the state space is:*

$$\Omega_{2}=\left\{\mathrm{Asn},\;\mathrm{Asp},\;\mathrm{Arg},\;\mathrm{Cys},\;\mathrm{Leu},\;\mathrm{Lys},\;\mathrm{Gln},\;\mathrm{Ser},\;\mathrm{Val},\;\mathrm{STOP}\right\}.$$


*In the application of the method described below, it is important that in both tables the last element occurs at least twice. Assume that in the sequence in Table 4, the TCC codon (corresponding to the amino acid serine (Ser)) is missing. We want to properly complete the following sequence including three adjacent gaps:*

(8)
$$\begin{array}{c} \mathrm{GTCAAAAGACTCTCCGACAAAAACAAA}□□□\mathrm{AGTCTC}\\ \mathrm{TAGCAGTTATGTTGTTAG} \end{array}$$


*In the other approach, using representation with amino acids (see Table 3), we want to complete the following corresponding sequence including a single gap:*

(9)
ValLysArgLeuSerAspLysAsnLys□SerLeuSTOPGlnLeuCysCysSTOP.


*The (extensive) transition matrix corresponding to sequence (Equation 8) is:*

(10)
$$\Pi_{1}=\begin{array}{cc} \; & \begin{array}{cccc} A & \;C & \;G & \;T \end{array}\\ \begin{array}{c} \;A\\ \;C\\ \;G\\ \;\;T \end{array}\; & \;\left(\begin{array}{cccc} \frac{9}{18} & \frac{3}{18} & \frac{5}{18} & \frac{1}{18}\\ \frac{4}{10} & \frac{1}{10} & \frac{1}{10} & \frac{4}{8}\\ \frac{2}{8} & \frac{1}{8} & \frac{0}{8} & \frac{5}{8}\\ \frac{3}{13} & \frac{5}{13} & \frac{2}{13} & \frac{3}{13} \end{array}\right) \end{array}$$


*Let us observe that the number of occurrences of G in sequence (Equation 8) is 8, as we do not count the last occurrence, because it is not paired. The number of occurrences of A in sequence (Equation 8) is 18, as we do not count the last occurrence of A, before the lacking fragment, because this occurrence is not paired. Analogously, the transition matrix corresponding to sequence (Equation 9) is:*

(11)
$$\Pi_{2}=\begin{array}{cc} \; & \begin{array}{cccccccccc} \mathrm{Asn} & \mathrm{Asp} & \mathrm{Arg} & \mathrm{Cys} & \mathrm{Leu} & \mathrm{Lys} & \mathrm{Gln} & \mathrm{Ser} & \mathrm{Val} & \mathrm{Stop} \end{array}\\ \begin{array}{c} \mathrm{Asn}\\ \mathrm{Asp}\\ \mathrm{Arg}\\ \mathrm{Cys}\\ \mathrm{Leu}\\ \mathrm{Lys}\\ \mathrm{Gln}\\ \mathrm{Ser}\\ \mathrm{Val}\\ \mathrm{Stop} \end{array}\; & \;\left(\begin{array}{cccccccccc} \;0\; & \;0\; & \;0\; & \;0\; & \;0\; & \;1\; & \;0\; & \;0\; & \;0\; & \;0\;\\ 0 & 0 & 0 & 0 & 0 & 1 & 0 & 0 & 0 & 0\\ \frac{1}{3} & 0 & \frac{1}{3} & 0 & 0 & 0 & 0 & \frac{1}{3} & 0 & 0\\ 0 & 0 & 0 & \frac{1}{2} & 0 & 0 & 0 & 0 & 0 & \frac{1}{2}\\ 0 & 0 & 0 & \frac{1}{3} & 0 & 0 & 0 & \frac{1}{3} & 0 & \frac{1}{3}\\ \frac{1}{3} & 0 & \frac{1}{3} & 0 & 0 & 0 & 0 & \frac{1}{3} & 0 & 0\\ 0 & 0 & 0 & 0 & 1 & 0 & 0 & 0 & 0 & 0\\ 0 & \frac{1}{3} & 0 & 0 & \frac{1}{3} & 0 & 0 & \frac{1}{3} & 0 & 0\\ 0 & 0 & 0 & 0 & 0 & 1 & 0 & 0 & 0 & 0\\ 0 & 0 & 0 & 0 & 0 & 0 & 1 & 0 & 0 & 0 \end{array}\right) \end{array}$$


*Note that, after rounding, the stationary probability vector of matrix $\Pi_{1}$ is equal to:*

$$\pi_{1}\approx \left(0.367347,0.204082,0.163265,0.265306\right).$$


*For the sake of comparison, we below give rounded relative frequencies of nucleotide occurrences, as disclosed in Table 4:*

A:0.351852,C:0.222222,G:0.166667,T:0.259259.


*The values sourced from Table 4 are not identical to the respective coordinates of the vector $\pi_{1}$, given that:*
*1*. 
*the sequence of 54 nucleotides is short,*
*2*. 
*in sequence (Equation 8), three nucleotides are lacking (cf. Remark 14).*


*Using the transition matrix $\Pi_{1}$, we will run an experiment, described below, which will allow us to complete sequence (Equation 8). One can proceed analogously using the matrix $\Pi_{2}$, which we will not do, given the symmetry of the method. The description of the experiment makes it possible to repeat it with no IT tools.*

*Prepare four boxes, labeled A, C, G and T. In each box, there are assorted balls labeled A, C, G and T. The numbers of balls of individual colors in box A are in proportion to respective entries of the first row of the matrix $\Pi_{1}$. Thus there are 9 balls labeled A, 3 balls labeled C, 5 balls labeled G, and 1 ball labeled T, a total of 18 balls. We fill the other boxes (C, G and T) analogously. Now the experiment begins. In sequence (Equation 8), there is a gap after A. Therefore, we draw one ball from box A at random. Assume we have drawn T, which can be done with probability $\frac{1}{18}$. In the next step we draw from the box labeled the same way as the most recently drawn ball; in our experiment it is box T. Assume we have drawn C from box T, which can be done with probability $\frac{5}{13}.$ Proceeding this way, we now draw a ball from box C. Assume we have again drawn C, which can be done with probability $\frac{1}{10}.$ Thus we have generated three consecutive elements of the sequence GTCAAAAGACTCTCCGACAAAAACAAA, namely TCC, and we fill the gap in sequence (Equation 8) with this result. We have recovered the original sequence presented in Table 4. The algorithm works as shown in Figure 6.*

*For the reader’s convenience, we present in Figure 7 below a computer program written in the Python 3 language, the source code is also available in GitHub [20].*


## 5. Conclusions

The method just presented is primarily of educational value. The procedure is described suggestively and, the authors believe, explanatory, which makes it possible for the method to be used in a more general context, also by non-mathematicians. The authors do not imply that the probability of achieving the proper completion of a DNA genome is satisfactory; they only present a tool which may be used for such completion and with which they would like to familiarize the reader. The authors are aware that the contemporary efforts in the area of DNA genome completion are focusing on deep learning rather than on Markov chains even of higher orders, see [8]. This paper proposes an alternative tool that can be explained suggestively and deeply. It is now clear that the deep learning methods lead to more exact completions than the Markov chains methods. Yet, our method allows for understanding of what happens behind the process of proper completion and sequencing. It should therefore be treated as of explanatory and educational value, with a potential for future research. It is worth asking, if the algorithms presented in Example 2 might be used to easily generate test data for more advanced deep learning algorithms (see [21]).

## Figures and Tables

**Figure 1 genes-13-00554-f001:**
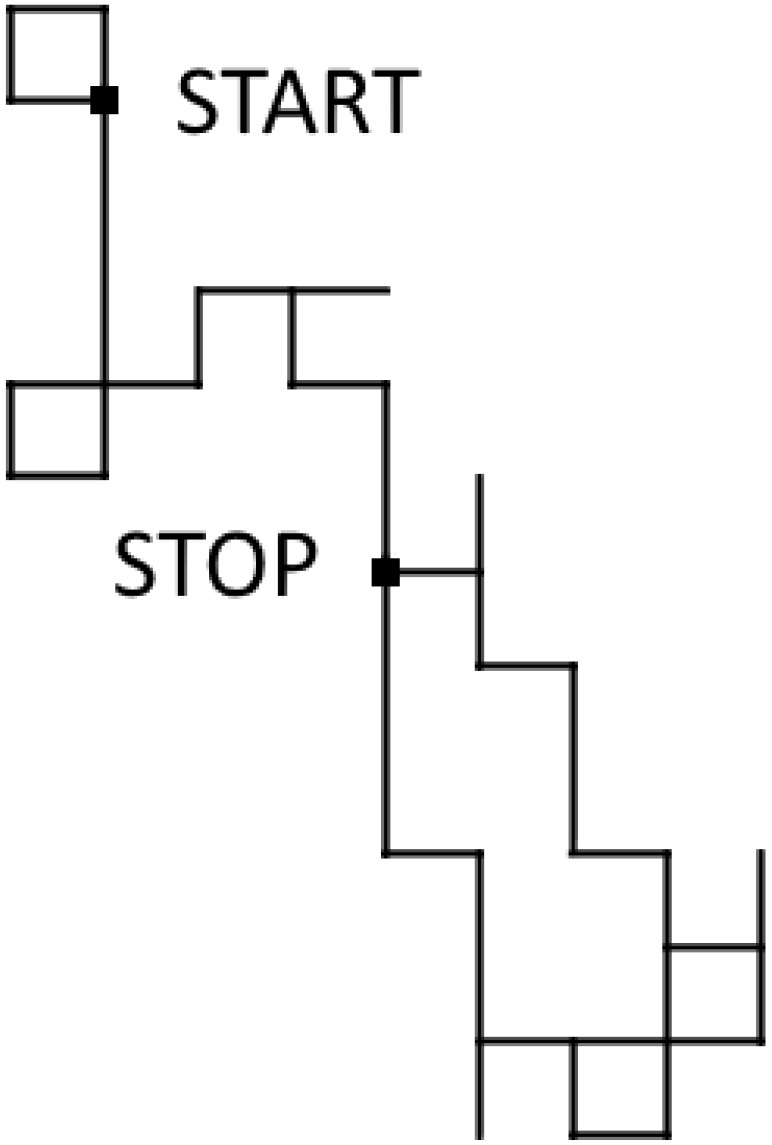
Fifty-four step random walk from a central point on a complex plane. Based on DNA sequence from Example 2.

**Figure 2 genes-13-00554-f002:**
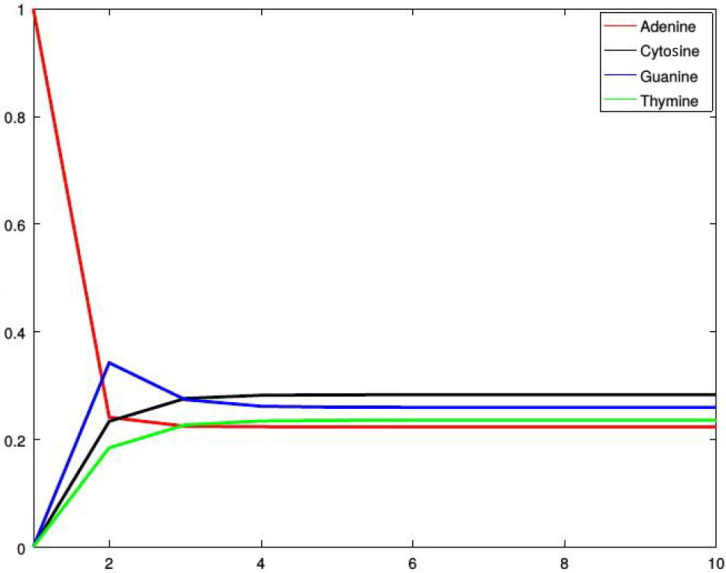
Probabilities of occurrences of the bases in consecutive steps starting from adenine.

**Figure 3 genes-13-00554-f003:**
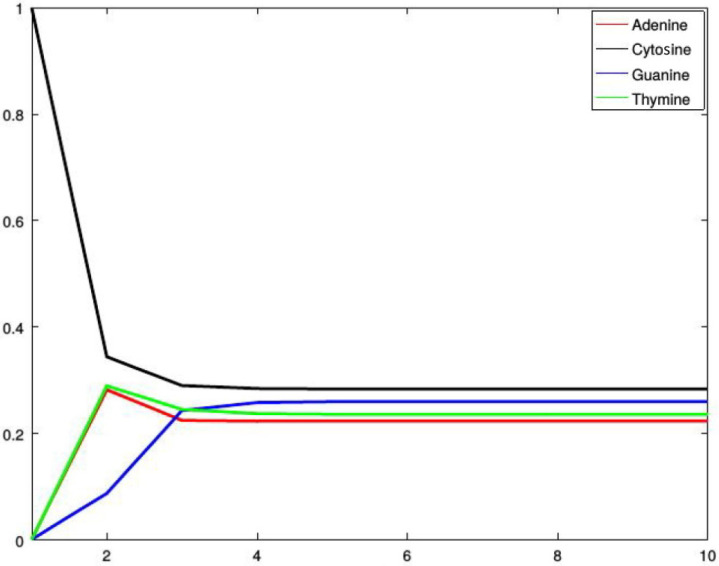
Probabilities of occurrences of the bases in consecutive steps starting from cytosine.

**Figure 4 genes-13-00554-f004:**
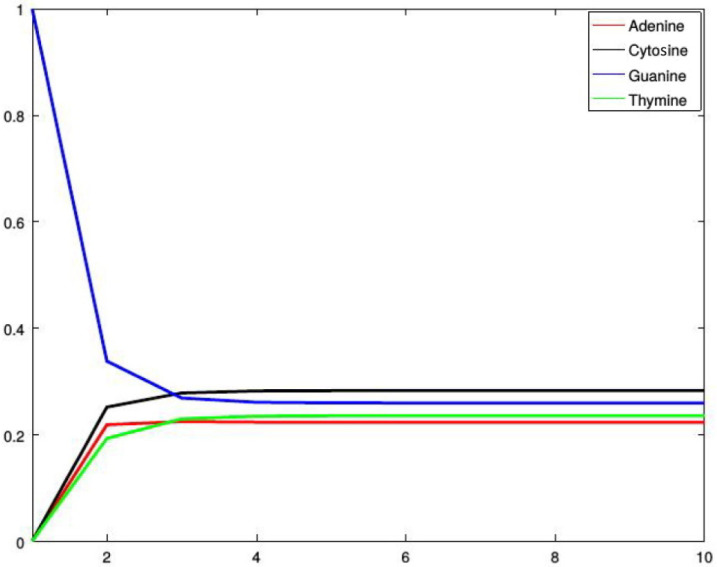
Probabilities of occurrences of the bases in consecutive steps starting from guanine.

**Figure 5 genes-13-00554-f005:**
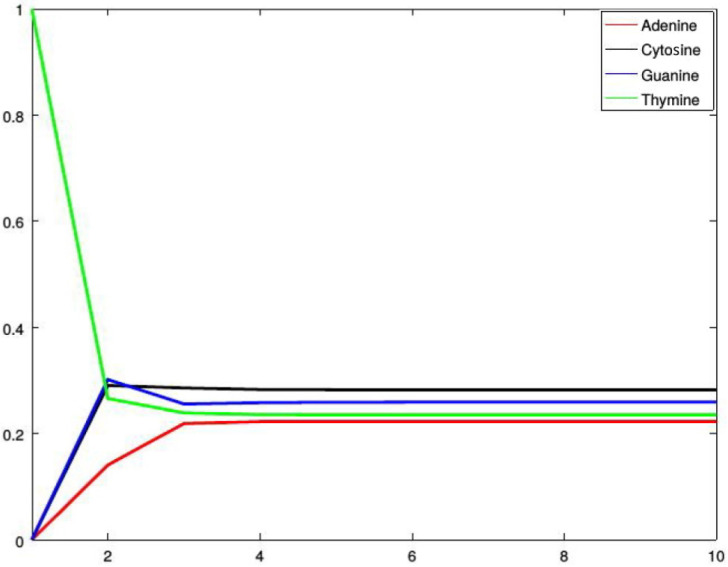
Probabilities of occurrences of the bases in consecutive steps starting from thymine.

**Figure 6 genes-13-00554-f006:**
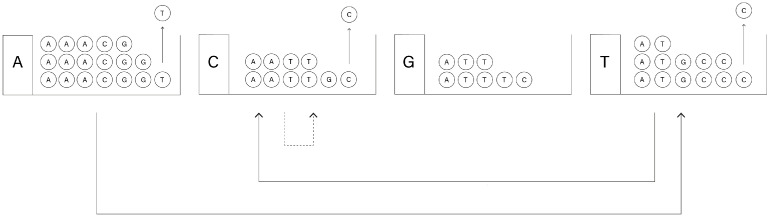
Illustration for Example 2.

**Figure 7 genes-13-00554-f007:**
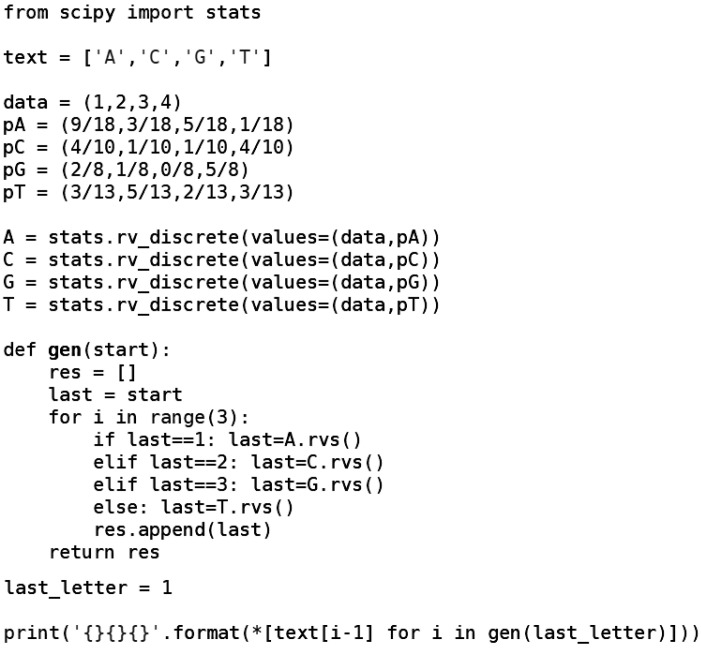
Program executes computations from Example 2 with probabilities from the transition matrix $\Pi_{1}$ and generates the completion of sequence (Equation 8) with exactly three elements.

**Table 1 genes-13-00554-t001:** Occurence of nucleotide bases in A3GALT2.

Nucleotide Bases	Occurence in A3GALT2	Probability (Occurence Divided by Length of Gene)
A	3195	0.222912
C	4052	0.282704
G	3715	0.259192
T	3371	0.235192

**Table 2 genes-13-00554-t002:** Occurence of nucleotide pairs of bases in A3GALT2.

Pairs of Bases	Occurence in A3GALT2	Probability	Pairs of Bases	Occurence in A3GALT2	Probability
AA	769	$\frac{769}{14,332}$	GA	811	$\frac{811}{14,332}$
AC	745	$\frac{745}{14,332}$	GC	934	$\frac{934}{14,332}$
AG	1093	$\frac{1093}{14,332}$	GG	1254	$\frac{1254}{14,332}$
AT	588	$\frac{588}{14,332}$	GT	716	$\frac{716}{14,332}$
CA	1140	$\frac{1140}{14,332}$	TA	475	$\frac{475}{14,332}$
CC	1392	$\frac{1392}{14,332}$	TC	980	$\frac{980}{14,332}$
CG	350	$\frac{350}{14,332}$	TG	1018	$\frac{1018}{14,332}$
CT	1170	$\frac{1170}{14,332}$	TT	897	$\frac{897}{14,332}$

**Table 3 genes-13-00554-t003:** The sequence of amino acids corresponding to 54-base-pair-long segment of SATB1.

1,2,3	4,5,6	7,8,9	10,11,12	13,14,15	16,17,18	19,20,21	22,23,24	25,26,27
Val	Lys	Arg	Leu	Ser	Asp	Lys	Asn	Lys
28,29,30	31,32,33	34,35,36	37,38,39	40,41,42	43,44,45	46,47,48	49,50,51	52,53,54
Ser	Ser	Leu	STOP	Gln	Leu	Cys	Cys	STOP

**Table 4 genes-13-00554-t004:** Numbered 54-base-pair-long segment of SATB1.

1	2	3	4	5	6	7	8	9	10	11	12	13	14	15	16	17	18
G	T	C	A	A	A	A	G	A	C	T	C	T	C	C	G	A	C
19	20	21	22	23	24	25	26	27	28	29	30	31	32	33	34	35	36
A	A	A	A	A	C	A	A	A	T	C	C	A	G	T	C	T	C
37	38	39	40	41	42	43	44	45	46	47	48	49	50	51	52	53	54
T	A	G	C	A	G	T	T	A	T	G	T	T	G	T	T	A	G

## Data Availability

Not applicable.

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
