# Peer review of "Note on DNA Analysis and Redesigning Using Markov Chain"

_genes, 2022, doi:10.3390/genes13030554_

Round 1
Reviewer 1 Report
This study discusses DNA analysis and redesigning using Markov chain. I should appreciate the authors' time and patience to come up with some results. However, there are still several problems that deduct from the quality of this manuscript. Below are several comments on this work.
- How do you verify the DNA design results?
- In Conclusion, there was no mention of the limitations of the study.
- The authors should proofread the English writing to improve the study.
Reviewer 2 Report
The manuscript by Maciej Zakarczemny and Małgorzata Zajęcka nicely introduces the concept of Markov chain and its applicability for assessing the missing segments in DNA sequences. The authors have paid too much attention to explaining their concept and the underlying theory. Although I greatly appreciate and respect their work, I must be objective and point out to the authors that we no longer use such Markov chain-based models in the field of bioinformatics and computational biology because the comparison of real DNA sequences and missing sequences assessed by Markov chains does not fit perfectly. In particular, such comparison studies with high-ranking NGS sequences and predicted sequences have not provided satisfactory results. This is one of the main reasons why almost every bioinformatics group considers deep learning models for this type of analysis.
Some minor comments are:
- In Definition 1.1: I wonder why the authors do not refer to their matrix as a doubly stochastic matrix. There is a need for explanation here.
- Although the appropriate explanations for the figures (2,3,4,5) and tables (3 and 4) are given in the text. All figures and tables require clear and well explained labels.
Overall, as already pointed out by the authors, the manuscript is very well suited for teaching purposes. For example, in computational biology lectures it might be considered, but for publication it lacks novelty
Round 2
Reviewer 2 Report
The authors have been very responsive to all my comments, and the manuscript meets the requirements of a good scientific paper,